# BlockmiR AONs as Site-Specific Therapeutic MBNL Modulation in Myotonic Dystrophy 2D and 3D Muscle Cells and HSA^LR^ Mice

**DOI:** 10.3390/pharmaceutics15041118

**Published:** 2023-03-31

**Authors:** Sarah J. Overby, Estefanía Cerro-Herreros, Jorge Espinosa-Espinosa, Irene González-Martínez, Nerea Moreno, Juan M. Fernández-Costa, Jordina Balaguer-Trias, Javier Ramón-Azcón, Manuel Pérez-Alonso, Thorleif Møller, Beatriz Llamusí, Rubén Artero

**Affiliations:** 1University Institute of Biotechnology and Biomedicine (BIOTECMED), Universidad de Valencia, 46100 Burjassot, Spain; 2Human Translational Genomics Group, Incliva Biomedical Research Institute, 46010 Valencia, Spain; 3Institute for Bioengineering of Catalonia (IBEC), The Barcelona Institute of Science and Technology, Baldiri I Reixac 10–12, 08028 Barcelona, Spain; 4ICREA—Institució Catalana de Recerca i Estudis Avançats, 08010 Barcelona, Spain; 5Ranger Biotechnologies A/S, Skovvænget 32, 5792 Aarslev, Denmark

**Keywords:** Myotonic Dystrophy 1, MBNL, muscleblind, antisense oligonucleotides, AON, miRNA, blockmiR, miR-23b, miR-218

## Abstract

The symptoms of Myotonic Dystrophy Type 1 (DM1) are multi-systemic and life-threatening. The neuromuscular disorder is rooted in a non-coding CTG microsatellite expansion in the DM1 protein kinase (*DMPK*) gene that, upon transcription, physically sequesters the Muscleblind-like (MBNL) family of splicing regulator proteins. The high-affinity binding occurring between the proteins and the repetitions disallow MBNL proteins from performing their post-transcriptional splicing regulation leading to downstream molecular effects directly related to disease symptoms such as myotonia and muscle weakness. In this study, we build on previously demonstrated evidence showing that the silencing of miRNA-23b and miRNA-218 can increase MBNL1 protein in DM1 cells and mice. Here, we use blockmiR antisense technology in DM1 muscle cells, 3D mouse-derived muscle tissue, and in vivo mice to block the binding sites of these microRNAs in order to increase *MBNL* translation into protein without binding to microRNAs. The blockmiRs show therapeutic effects with the rescue of mis-splicing, MBNL subcellular localization, and highly specific transcriptomic expression. The blockmiRs are well tolerated in 3D mouse skeletal tissue inducing no immune response. In vivo, a candidate blockmiR also increases Mbnl1/2 protein and rescues grip strength, splicing, and histological phenotypes.

## 1. Introduction

Myotonic Dystrophy Type 1 (DM1) is a multi-systemic neuromuscular disorder with onsets ranging from congenital to late adult. The disease affects the muscles causing symptoms including myotonia and muscle weakness but can also cause symptoms such as cardiac arrhythmias, insulin resistance, and cataracts (OMIM #160900). DM1 affects around 4.76 people per 10,000 [1]. Patients report that muscle weakness/locking and fatigue add difficulty to their daily lives, as well as emotional and interpersonal strain [2]. More recently, it was shown that DM1 patients are more prone to contracting severe COVID-19 with poor outcome due to respiratory muscle weakness [3]. Currently, there is no disease-targeting treatment available. However, as of 2022, approximately 20 drug candidates are in the preclinical phase of the development pipeline with 3 already in clinical trial [4].

DM1 is a microsatellite expansion disease caused by toxic “CTG” repetitions in the DM1 Protein Kinase (*DMPK*) gene. Signs of pathogenicity begin with the presence of 50 or more repetitions and commonly reach into the hundreds, or even thousands in the most severe cases [5]. The repetitions bind and sequester the Muscleblind-like (MBNL) family of proteins into nuclear RNA foci causing MBNL loss-of-function [6]. The family is made up of MBNL1, MBNL2, and MBNL3, all of which are sequestered by *DMPK* mRNA [7]. MBNLs are important splicing and polyadenylation site regulators and cause a shift from fetal to adult splicing patterns during development [8,9,10,11]. While MBNLs are repressed, the CUGBP Elav-Like Family Member 1 (CELF) family of proteins, which are also post-transcriptional splicing regulators, are activated due to their antagonistic relationship with the Muscleblind-like proteins [12]. MBNL sequestration and CELF activation cause a cascade of mis-splicing events resulting in an abnormal fetal pattern of many transcripts [13,14]. For example, *CLCN1* [15] and *INSR* [16] mis-splicing are responsible for myotonia and insulin resistance, respectively, while *SCN5A* is linked to heart conduction defects and cardiac arrhythmias [17].

Antisense oligonucleotides (AONs) have been used as a treatment strategy for this disease by targeting the toxic repetitions [18,19]. Specifically, AOC 1001 is currently in Phase1/2 clinical trial for DM1 therapy (ClinicalTrials.gov Identifier: NCT05027269). Indeed, AONs are a promising nucleic acid therapy across a variety of neuromuscular disorders [20]. The upregulation of MBNL1 has also been shown in several studies to be a valid therapeutic approach for DM1. For example, calcitriol, used to increase MBNL1 in DM1 mice, was able to strengthen mouse grip and rescue splicing [21].

Recently, microRNAs (miRNAs) have also been shown to contribute to the phenotype seen in DM1. Specifically, miR-218 was shown to be upregulated in DM1 patient muscle biopsies [22]. miR-218 and miR-23b are known regulators of *MBNL* transcripts through specific 3′-untranslated region (3′-UTR) binding. Targeting of these miRNAs with the use of antagomiR AONs has shown rescue of MBNL1 protein level in DM1 cells [23] as well as in HSA^LR^ mice [24]. Other rescues observed in these studies include alternative splicing, histological phenotypes, myotonia, and grip strength. The roles of other miRNAs like miR-7 [25], miR-1 [26], and miR-30 [27] are also being investigated for their potential implications in DM1.

The use of blockmiRs is an alternative strategy to direct miRNA inhibition. BlockmiRs specifically bind to a 3′-UTR site on the target transcript instead of binding to miRNAs. This blocks the interaction of miRNAs to the 3′-UTR and allows translation to occur. This strategy has already shown success in our previous study using peptide-linked morpholino (P-PMO) technology [28,29]. In that study, the P-PMO blockmiR showed proof of concept by increasing Mbnl1 protein levels and functionality evidenced through improvements of grip strength, mis-splicing, and histological phenotypes. The blockmiR also showed highly specific effects through miRNA site blocking as several disease-unrelated miR-23b targets were unaffected by the treatment.

Our previous study only showed the effects of a blockmiR with P-PMO chemistry at one of the binding sites for miR-23b in immortalized patient-derived DM1 fibroblasts transdifferentiated into myotubes (DM1 cells). Here, we build on the previous study by designing blockmiRs with locked nucleic acid (LNA)-based chemistry against multiple binding sites of miR-23b and miR-218 on *MBNL1* and *MBNL2* 3′-UTRs and tested in DM1 cells as well as mouse-derived 3D skeletal muscle tissue [30,31]. Importantly, we also demonstrate that the candidate blockmiR continues to show therapeutic effects on the DM1 cell transcriptome in a site-specific manner using RNA-Seq. BlockmiR treatment increased the amount of MBNL1 transcripts and protein and rescued splicing patterns in these cells. A candidate compound was also administered in HSA^LR^ mice [32] increasing grip strength and Mbnl1 protein as well as improving histological phenotypes and aberrant splicing.

## 2. Materials and Methods

### 2.1. Cell Culture

Healthy immortalized control (CNT) [33] and patient-derived (DM1) fibroblasts carrying 1300 CTG repeats were provided by Dr. Furling (Institute of Myology, Paris, France) and were transdifferentiated into myotubes by inducing MyoD expression. The fibroblasts were grown in Dulbecco’s Modified Eagle Medium (DMEM, 4.5 g/L glucose, Gibco), 1% penicillin and streptomycin (P/S, 10,000 U/mL, Thermo Fisher, Waltham, MA, USA), and 10% fetal bovine serum (FBS, Sigma-Aldrich, Saint Louis, MI, USA). Transdifferentiation was prompted by muscle differentiation medium (MDM) containing DMEM with 4.5 g/L glucose, 1% P/S, 2% horse serum, 1% apo-transferrin (10 mg/mL), 0.1% insulin (10 mg/mL), and 0.02% doxycycline (10 mg/mL). Murine C2C12 skeletal myoblasts, purchased from the American Type Culture Collection (CRL-1772, ATCC, Manassas, VA, USA), were grown in DMEM (4.5 g/L glucose, Gibco, Thermo Fisher, Waltham, MA, USA ) supplemented with 1% P/S (10,000 U/mL, Thermo Fisher, Waltham, MA, USA) and 10% FBS (Gibco, Thermo Fisher, Waltham, MA, USA). All cells were grown at 37 °C in a humidified atmosphere containing 5% CO_2_.

### 2.2. Mouse 3D Skeletal Muscle Tissues Biofabrication

Cell-laden 3D micro-structured GelMA-CMCMA composite hydrogels were fabricated on top of 6-well plate transwells using a photo-mold patterning technique, as previously described [30]. For this, the cells used were immortalized human fibroblasts from an unaffected control and a DM1 patient (carrying 1300 CTG repeats quantified in the blood cells), which were provided by Dr. Denis Furling and Dr. Vincent Mouly (Institute of Myology, Paris, France). To create the 3D structure, prepolymer solution was mixed with a cell suspension in growth medium to obtain a final concentration of 2.5 × 10^7^ cells/mL. Then an 8 µL drop of the cell-laden prepolymer was placed on the membrane of the transwell and a PDMS stamp was pressed lightly on top, filling the microchannels with the solution. The hydrogels were crosslinked by 30 s of UV exposure using the UVP Crosslinker. Growth medium was added to each sample and stamps were carefully removed after 20 min of incubation at 37 °C. Two days after cell encapsulation, the medium was replaced with a differentiation medium (DMEM (4.5 g/L glucose (Gibco, Thermo Fisher, Waltham, MA, USA) supplemented with 1% P/S (10,000 U/mL, Thermo Fisher) and 2% horse serum (HS, Gibco, Thermo Fisher, Waltham, MA, USA) to induce myotube maturation. Bioengineered 3D skeletal muscle tissues were differentiated for up to 10 days with culture media being replaced every two days. Representative images of the 3D tissue can be found in Appendix A and other examples can be found in a previous publication [31].

### 2.3. BlockmiR Design

BlockmiRs were designed as nucleic acid duplexes with a 2′-O-methyl core and LNA-modified ends with phosphorothioate linkages (Table 1). The lowercase “a, g, c, u” are for 2′-O-methyl nucleotides, “Ab, Gb, Tb, Cb” are for LNA nucleotides, and “s” are for phosphorothioate linkages. A scrambled oligo (Scramble) without predicted specific binding was also designed as a control for the 2′-O-methyl/LNA chemistry. A more detailed description of all blockmiR sequences and their modifications can be found in Appendix A.

### 2.4. Cell Transfection

DM1 and CNT fibroblasts were plated at 1.0 × 10^5^ cells per mL in 6 cm Petri dishes (Falcon, VWR, Radnor, PA, USA). Transfection concentrations for RNA extractions as well as immunofluorescence can be found in Table 1. The Scramble was transfected at 50 nM and 200 nM to match the concentrations of the blockmiRs. Cell transfection was carried out with XtremeGENE™ HP (Roche, Penzberg, Germany) and treated with MDM to promote transdifferentiation. After 4 days, RNA was extracted from the cells.

### 2.5. Toxicity Assay

DM1 cells were plated at 1.0 × 10^5^ cells per mL in a 96-well plate (Falcon, VWR, Radnor, PA, USA). After 24 h, blockmiRs were transfected at increasing concentrations (2 nM, 10 nM, 50 nM, 200 nM, 1 μM, and 5 μM). These were performed in triplicate and left for four days using MDM to promote transdifferentiation into myotubes. Cell viability was measured by adding 20 μL of MTS/PMS tetrazolium salt from the CellTiter 96 Aqueous Non-Radioactive Cell Proliferation Assay (Promega, Madison, WI, USA) to each well and incubating for four hours at 37 °C in a humidified incubator with 5% CO_2_. As MTS is converted into soluble formazan, the absorbance was measured at 490 nm using an Infinite 200 PRO plate reader (Tecan, Maennedorf, Switzerland). Data from the transfected cells were calculated with the following formula:$$100-\frac{(\mathrm{transfected}\;\mathrm{cells}-\mathrm{mean}\;\mathrm{only}\;\mathrm{media}\;\mathrm{control})\times 100}{\mathrm{mean}\;\mathrm{non-transfected}\;\mathrm{cells}\;\mathrm{control}}$$

After normalization, data were log-transformed and the threshold concentration values at 50% (TC_50_) were calculated with the least squares non-linear regression model.

### 2.6. MagnetoELISA

To evaluate the innate immune activation by the blockmiRs, the secretion of interleukin-6 (IL-6) was analyzed in the C2C12-derived 3D mouse skeletal muscle tissues after treatment with blockmiRs using a custom MagnetoELISA assay. Murine C2C12 skeletal myoblasts were purchased from the American Type Culture Collection (CRL-1772, ATCC, Manassas, VA, USA). The assays were performed in engineered 3D skeletal muscle tissues with an internal positive control using Lipopolysaccharide (LPS) at 10 µg/mL (three replicates per assay). The dose dependency of IL-6 upon LPS exposure has been shown previously [34]. Negative control was used using phosphate-buffered saline (PBS) (three replicates per assay). The blockmiRs were tested in each assay at 500 nM of concentration (three replicates each). A 12-point calibration curve was performed at well-known concentrations of the respective cytokine (Appendix A).

The IL-6 antibody was coupled covalently to the magnetic beads as previously described [34]. Magneto ELISAs optimized assay was performed using transwell permeable supports as a platform to develop the immune-sensing measurement procedure. Forty-eight hours after blockmiR administration to 3D skeletal muscle culture, just before chemical stimulation, antibody-bioconjugated magnetic microbead (MB·Abs) (5 µg for IL-6) were loaded in the lower side of the transwell, where the capture step was performed. Samples were incubated for 1 h under soft stirring to avoid bead aggregation inside an incubator at 37 °C and 5% CO_2_ atmosphere. During the experiments, the upper and lower part of the transwell was filled with 1 mL and 2 mL of cell medium, respectively. Secreted cytokines, as a result of blockmiR administration, diffused through the transwell membrane and were captured by the IL-6 antibodies immobilized onto the surface of the beads. After this time, the MB·Abs were washed in PBST-BSA buffer (3X) and a biotinylated antibody was added at 1 µg/mL. After 30 min under stirring, the MB·Abs were washed again and resuspended in SAV-pHRP 0.50 µg/mL (for 30 min under the same conditions described before). Afterward, the MB·Abs were washed and resuspended in 100 µL of citrate buffer-KCl and captured onto the surface of the microplates using a custom-made polymethyl methacrylate (PMMA) cell with magnets. Finally, absorbance measurements were performed at 450 nm.

### 2.7. RNA Extraction, RT-qPCR, and RT-PCR

Total RNA from transdifferentiated DM1 myotubes (see Section 2.1 Cell culture) or mouse quadriceps and gastrocnemius muscle (see Section 2.12. Transgenic mice) was isolated using QIAzol Lysis Reagent and RNeasy Mini Kit (Qiagen, Hilden, Germany) according to the manufacturer’s instructions. One microgram of RNA was digested with DNase (Zymo, Irvine, CA, USA) and reverse-transcribed with SuperScript II Reverse Transcriptase (Invitrogen, Thermo Fisher, Waltham, MA, USA) using random hexanucleotides (Roche, Penzberg, Germany). Replicates of cDNA were generated in three independent experiments.

Real-time (RT) PCR was performed using approximately 4 ng of template cDNA with TaqMan probes (Qiagen, Hilden, Germany) for *MBNL1* and *MBNL2* with FAM-labeled probes and GAPDH as an endogenous control with MAX-labeled probes (Appendix A). This was performed similarly for mouse tissue RNA extractions while using mouse-specific Mbnl1, Mbnl2, and Gapdh probes. Three technical replicates were performed for each sample. Expression levels were measured using an Applied Biosystems StepOnePlus Real-Time PCR System. Expression relative to the endogenous gene and calibrated with the Scramble control group was calculated using the 2^−ΔΔCt^ method.

Alternative splicing was analyzed using approximately 200 ng of cDNA in a standard semiquantitative PCR reaction with GoTaq polymerase (Promega, Madison, WI, USA) and specific primers for *MBNL1*, *NFIX*, and *SPTAN1*, with *GAPDH* as an endogenous control (Appendix A). Mouse tissue RNA was analyzed similarly with primers for *Atp2a1*, *Clcn1*, *Mbnl1*, *Nfix*, and *Gapdh*. PCR products were run on a 2% agarose gel and bands quantified with Image J software (NIH, Version 1.53a) and compared to Scramble control cells. Full-length gels can be found in Appendix A.

### 2.8. Immunofluorescence

In a 24-well plate (Falcon, VWR, Radnor, PA, USA), DM1 cells were seeded at 4.0 × 10^4^ cells per well. They were then transfected and left for four days in MDM for transdifferentiation. A 4% paraformaldehyde (PFA) solution was added to the transdifferentiated myotubes for 15 min at room temperature (RT) in order to fix the cells and then the cells were washed with PBS three times. PBS-T (0.3% Triton-X in PBS) was used to permeabilize the myotubes and then blocked (PBS-T, 0.5% BSA, 1% donkey serum) for 30 min at RT. MBNL1 primary antibody (1:200, MB1a (4A8), Developmental Studies Hybridoma Bank, Iowa City, IA, USA [35]) was used for overnight incubation at 4°C. The cells were then washed three times with PBS-T and incubated for 1 h with a biotinylated anti-mouse-IgG (1:200, Sigma-Aldrich, Saint Louis, MI, USA) with later Avidin-Biotin amplification (Elite ABC kit, VECTASTAIN, Vector Laboratories, Newark, CA, USA) for 30 min at RT. This was followed by three PBS-T washes and incubation with streptavidin-FITC fluorophore (1:200, Vector Laboratories, Newark, CA, USA) for 2 h at RT. After three more washes with PBS, the slides of cells were mounted with VECTASHIELD mounting medium containing 2 μg/mL DAPI (Vector Laboratories, Newark, CA, USA) for nuclei visualization.

An Olympus FluoView FV100 confocal microscope at 400x magnification was used for imaging the myoblasts. The images were quantified using Image J (NIH, Version 1.53a) with a threshold of 10 using the following formula:$$\mathrm{Mean}\;\mathrm{Pixel}\;\mathrm{Intensity}=\frac{\mathrm{Gray}\;\mathrm{value}}{\mathrm{Area}}$$

### 2.9. Protein Extraction and Analysis

For the activity assay, cells were seeded in 6-well plates at a density of 8.0 × 10^4^ cells per well and transfected 24 h later with blockmiRs, as previously explained. For total protein extraction, human DM1 myotubes (see Section 2.1 Cell culture) or mouse quadriceps and gastrocnemius (see Section 2.12. Transgenic mice) were sonicated and homogenized in Pierce RIPA buffer (Thermo Fisher, Waltham, MA, USA) supplemented with protease and phosphatase inhibitor cocktails (Roche, Penzberg, Germany). Quantification of total protein was performed with a BCA protein assay kit (Pierce, Thermo Fisher, Waltham, MA, USA) using bovine serum albumin as a standard. For the immunodetection assay in the cells, the protocol was performed as described previously [36]. Briefly, 1 μg/well of cell samples was denatured (100 °C for 5 min) and loaded in quantitative dot blot (QDB) plates (Quanticision Diagnostics Inc, Research Triangle Park, NC, USA). Each cell sample was loaded at 2 μg/well in quadruplicate in two different plates; one was used to detect MBNL1 (1:1000, (4A8), Developmental Studies Hybridoma Bank, Iowa City, IA, USA [35]) and the other for GAPDH (1:500, (G-9) Santa Cruz Biotechnology, Dallas, TX, USA), which was used here as an endogenous control.

For the immunodetection assay in mouse tissue, the Jess Simple Western system was used to quantify Mbnl1 by chemiluminescence using anti-MBNL1 (1:10, ab45899, anti-rabbit, Abcam, Cambridge, UK) and normalized to total protein fluorescence without the Replex step. Full-length blots can be found in Appendix A. Mbnl2 protein was quantified by quantitative sandwich ELISA, according to the manufacturer’s instructions (MyBioSource, San Diego, CA, USA). Briefly, 20–40 mg of muscle was homogenized in 200 μL of 1 × PBS buffer (8 mM Na_2_HPO_4_, 150 mM NaCl, 2 mM KH_2_PO_4_, 3 mM KCl). Quantification of total protein was performed with a BCA protein assay kit (Pierce, Thermo Fisher, Waltham, MA, USA) using bovine serum albumin as standard. 20 μg was used for loading each sample.

### 2.10. RNASeq Transcriptome Analysis

Using 6 cm Petri dishes (Falcon, VWR, Radnor, PA, USA) DM1 cells were plated at 1.0 × 10^5^ cells per mL. The cells were allowed to transdifferentiate for four days using MDM. Then M1-Block-23b-1 and AntagomiR-23b were both transfected at a concentration of 50 nM and left for four more days of differentiation. Total RNA was extracted following the previously described extraction protocol on the eighth day. The TruSeq Stranded mRNA library preparation kit was used to generate libraries following Illumina protocols and these were sequenced using paired-end NextSeq 550 at a depth of 20 million reads per sample. Trim Galore! (RRID:SCR_011847, version 0.6.4_dev) software was used to filter out reads with a q-value below 30. Using STAR (version 2.7.3a) software, all accepted reads were aligned to the genome GRCh38.p12. The resulting BAM files were analyzed using RSEM (version v1.3.2) software to obtain gene counts. Differential gene expression (DGE) was calculated using R package edgeR (version 3.28.1). The threshold for a DGE call was an adjusted p-value < 0.05 and log2FoldChange of 1. Gene expression recovery, i.e., genes that approached a normal expression pattern, was performed for all disease-related genes using this formula:$$\%\;\mathrm{Recovery}\;-\frac{\mathrm{mean}\;\mathrm{treated}\;\mathrm{gene}\;\mathrm{counts}\;-\;\mathrm{mean}\;\mathrm{disease}\;\mathrm{gene}\;\mathrm{counts}}{\mathrm{mean}\;\mathrm{control}\;\mathrm{gene}\;\mathrm{counts}\;-\;\mathrm{mean}\;\mathrm{disease}\;\mathrm{gene}\;\mathrm{counts}}$$

DM1 muscle biopsy RNASeq data was retrieved from http://DMSeq.org (accessed on 13 November 2020. Splicing event analysis was performed using Vast-tools [37] with vast-tools merge for each group of samples to obtain the necessary read count, vast-tools compare with the flags –p_IR, –min_PSI 25 and –noVLOW. Next, all results were grouped and analyzed using in-house R scripts.

### 2.11. STRING Analysis

Genes of interest were investigated using the STRING Protein-Protein Interaction Network database [38]. The search filters included a minimum required interaction score of medium confidence (0.400) and the sources included textmining, experiments, databases, co-expression, neighborhood, gene fusion, and co-occurrence. Disconnected nodes were hidden from the network.

### 2.12. Transgenic Mice and BlockmiR Administration

Mouse handling and experimental procedures conformed to the European law regarding laboratory animal care and experimentation (2003/65/CE) and were approved by the Conselleria de Agricultura, Generalitat Valenciana (“Respuesta terapéutica a blockmiRs modificados en un modelo de ratón de DM1”; reference number 2020/VSC/PEA/0203). Homozygous transgenic HSA^LR^ (line 20 b) mice [32] were provided by Prof. C. Thornton (University of Rochester Medical Center, Rochester, NY, USA). FVB mice, obtained from Charles River Laboratories International, Inc. (Sulzfeld, Germany), were used as controls. Age-matched HSA^LR^ (∼3.5 months old) mice received one interscapular subcutaneous injection of 150 μL of either 1× PBS (*n* = 5), M1-Block23b-H 12.5 mg/kg (*n* = 5), or M1-Block23b-M 12.5 mg/kg (*n* = 5). FVB mice of the same age also received one interscapular subcutaneous injection of 150 μL of 1× PBS (*n* = 5). Four days after injection, the mice were sacrificed, and blood, muscles, and organs of interest were extracted. Muscles were cut in two with one half frozen in liquid nitrogen and the other half fixed with −80 °C chilled isopentane for histological processing.

### 2.13. Forelimb Grip Strength Test

Grip Strength Dynamometer (BIO-GS3; Bioseb, Pinellas Park, FL, USA) was used to test the forelimb grip strength of the mice. When the mouse grasped the bar, the peak pull force (grams) was recorded on a digital force transducer and then reset after each measurement. The tension recorded by the gauge was at the time the mouse released its forepaws from the bar. This was performed three consecutive times at 30 s intervals after body weight measurement. The percentage of normal force was calculated by normalizing the average strength after treatment to the strength before treatment and dividing this value by the body weight of each mouse.

### 2.14. Blood Biochemistry

The blood serum collected for each mouse was analyzed for biochemical parameters by Laboratorios Montoro Botella (Valencia, Spain). All samples were statistically compared with PBS mice. Kruskal–Wallis ANOVA was used and Wilcoxon statistical tests, when applicable.

### 2.15. Muscle Histology

The mouse quadriceps and gastrocnemius muscles frozen after fixation with −80 °C chilled isopentane were cut into 10-mm sections and stained with hematoxylin and eosin (H&E). The sections were mounted with VECTASHIELD mounting medium (Vector Laboratories, Newark, CA, USA) and images were taken with a Leica DM2500 microscope at 100× magnification. The percentage of fibers containing central nuclei was quantified in an average of 500 fibers in each mouse. All images were given random labels during quantification to facilitate a blind analysis.

### 2.16. Statistical Analysis

The calibration curve fittings, interpolation, and statistical analysis were performed using GraphPad Prism 8.2.1. (GraphPad Software Inc., San Diego, CA, USA).

## 3. Results

### 3.1. Preliminary Screen for blockmiRs in DM1 Cells

TargetScan and miRanda were used to predict the binding sites of miR-23b and miR-218 on the 3′-UTRs of *MBNL1* and *MBNL2*. They were then experimentally confirmed using a dual luciferase assay in HeLa cells in a previous study [23]. Three binding sites were confirmed on the *MBNL2* 3′-UTR for miR-218. One binding site was confirmed in the *MBNL1* 3′-UTR for miR-23b and one site in the *MBNL2* 3′-UTR. BlockmiRs were designed for all predicted and confirmed sites with LNA and 2′-*O*-methyl nucleotide modifications (Appendix A). However, only blockmiRs for the confirmed binding sites are shown in the following results (Figure 1a). The results of treatment with blockmiRs targeting unconfirmed binding sites can be found in Appendix A. A scrambled oligo (Scramble) was designed as a non-specific binding control for the AON chemistry.

The blockmiRs were first tested for toxicity after transfection into immortalized patient-derived fibroblasts transdifferentiated into myotubes (DM1 cells) [33] after four days at increasing concentrations to determine the threshold concentration level at 50% (TC50) of each treatment (Figure 1b). M2-Block-218-1 and the Scramble showed the lowest toxicity of the blockmiRs, which may suggest the absence of specific binding. The growth inhibition of all the blockmiRs was below the TC50 even at 200 nM concentration. Previously [23], therapeutic results were observed when antagomiRs targeting miR-23b were administered at 50 nM and those targeting miR-218 were administered at 200 nM in the same cells. Therefore, the same respective concentrations were chosen for blockmiRs targeting miR-23b or miR-218 so that the two drug strategies could be compared.

To test if blockmiRs activated an immune response, the effects of the oligos were quantified in a custom interleukin-6 (IL-6) MagnetoELISA assay. IL-6 is secreted in response to inflammation and immunity and is routinely studied as an indicator of stress [39]. The assays were performed in engineered 3D skeletal muscle tissues derived from mouse C2C12 cells. The 3D tissue was treated with lipopolysaccharide (LPS), which is a positive control for cytokine stimulation in skeletal muscle [40], and the blockmiRs at 500 nM. After treatment, the blockmiRs caused no increase in IL-6 secretion (Figure 1c).

### 3.2. Blocking miRNA Binding-Sites Shows Therapeutic Effects at the Transcript and Protein Level

The effects of blockmiR treatment at the molecular level were analyzed in DM1 cells four days after transfection and differentiation. A scrambled control with no predicted targets was used as a control at 50 nM (Scramble 50) and 200 nM (Scramble 200). BlockmiRs M1-Block-23b-1 and M2-Block-23b-1 both significantly increased the relative expression of *MBNL1* and *MBNL2* transcripts compared to the Scramble control (Figure 2a,b). M2-Block-23b-1 had a stronger effect on *MBNL2* transcripts, which is to be expected since this blockmiR targets the *MBNL2* 3′-UTR binding site. At the same time, blockmiRs M2-Block-218-2 and -3 both decreased the expression of *MBNL1* compared to Scramble 218 while leaving *MBNL2* transcripts unchanged.

The transcripts *MBNL1*, *NFIX*, and *SPTAN1* show abnormal splicing patterns in DM1 cells, which can be seen compared to control cells (CNT) (Figure 2c). Again, blockmiRs M1-Block-23b-1 and M2-Block-23b-1 showed completely reversed inclusion of exon 5 in *MBNL1* transcripts. The miR-218 blockmiRs had no effect on MBNL1 exon 5 or *NFIX* exon 7. However, they did show robust rescue of the inclusion of exon 23 in *SPTAN1* transcripts. The blockmiRs targeting miR-23b binding sites also slightly rescued *SPTAN1* exon 23 expression. MBNL1 proteins are known regulators of *MBNL1* and *NFIX* mRNAs [41], while MBNL2 regulates *SPTAN1* transcripts [42]. This may explain the stronger rescue of *SPTAN1* seen by blockmiRs targeting *MBNL2* miRNA binding sites. However, effects related to the miRNA target cannot be ruled out.

The goal of blockmiR treatment is to increase the translation of MBNL proteins by blocking the binding sites of miR-23b or miR-218. Thus, MBNL1 protein levels in DM1 cells were qualitatively and quantitatively analyzed through immunofluorescence staining after being treated with the blockmiRs. Representative images can be seen with MBNL1 protein in green and nuclear DAPI in blue (Figure 2d–i). In control cells, the MBNL1 signal is intense and disperses throughout the nucleus and the cytoplasm, indicating MBNL1 expression is healthy. In contrast, DM1 cells show a weak MBNL1 signal in the cytoplasm and with only small foci in the nucleus. The mean pixel intensity of MBNL1 fluorescence was calculated for each cell and normalized to cell area (Figure 2j). Like *MBNL1* transcripts, M1-Block-23b-1 and M2-Block-23b-2 both significantly increased MBNL1 protein fluorescence. These treatments also provoked a diffuse fluorescence throughout the cytoplasm indicating that the additional MBNL1 was not sequestered by the nuclear foci. Curiously, M2-Block-218-2 was also capable of increasing MBNL1 fluorescence pixel intensity. The Scramble 50 and 200 treatments caused no change in MBNL1 fluorescence. Images showing rescue after blockmiR treatment are shown in Figure 2 while the remaining photos can be found in Appendix A.

Looking at all the preliminary data, the most promising and consistent rescue was seen in DM1 cells treated with M1-Block-23b-1 or M2-Block-23b-1. These compounds both rescued the splicing patterns of MBNL1 and SPTAN1, while also significantly increasing MBNL1 transcript and protein. They also showed toxicity levels below the TC50 at 200 nM as well as no IL-6 immune response at high concentrations of 500 nM, while still inducing phenotypic improvements in the DM1 cells.

Considering future in vivo experiments, human *MBNL1* transcripts show more homology to mice than *MBNL2*. Specifically, miR-23b has no predicted binding sites on the mouse *Mbnl2* 3′-UTR. Thus, M1-Block-23b-1 was selected as the candidate blockmiR for deeper investigation and optimization. A dose-response study was conducted at five increasing concentrations of the blockmiR in DM1 cells and examined for MBNL1 protein expression through quantitative dot blot (QDB) (Figure 2k). A dose-responsive increase in MBNL1 protein was seen, with the exception of the 10 nM treatment. The treatment at 50 nM was also accordant with the results seen in immunofluorescence staining. Indeed, even at 2 nM, M1-Block-23b-1 was able to increase MBNL1 two-fold.

### 3.3. Transcriptomic Analysis Reveals Improvements by M1-Block-23b-1

RNASeq was used to find if M1-Block-23b-1 was capable of gene expression recovery before and after treatment. It was also used to compare the previously successful antagomiR counterpart and visualize the more global transcriptomic effects these oligos may have in DM1 cells. In this experiment, the immortalized fibroblasts were allowed to differentiate four days before transfection with the AONs and then left another four days in order to observe more prominent expression differences between the DM1 and healthy cells and to see a more analogous expression pattern to differentiated muscles in vivo.

First, splicing event alterations were identified between healthy and DM1 cells. Of the 478 events, these were quantified using a delta percent spliced-in (dPSI) formula. After applying the formula, events were narrowed to only those that had a dPSI over 25% and 10 or more reads, which identified 20 splicing events in 19 genes (Figure 3a). Of the 20 total events, 17 showed a reversal in splicing frequency after treatment with M1-Block-23b-1. This includes a splicing event in *SORBS1*, which is an MBNL1- and CELF1-regulated protein part of insulin signaling [43,44]. Another event was recovered in *MLH3*, which plays a role in the DNA mismatch repair pathway (MMR) and is connected to Huntington’s disease and the somatic expansion of CAG repeats [45,46]. Finally, *BAG4* showed splicing event recovery, which is a known target of miR-1 regulation [47]. This miRNA is variably expressed in DM1 tissues and likely regulates *DMPK* expression [48,49].

The second geometric point graph represents the top 25 genes showing total expression recovery as a result of treatment with M1-Block-23b-1 (Figure 3b). Because of this, the far-right column (CNT vs. BlockmiR) lacks geometric points since there is no difference in the levels of expression between control cells and DM1 cells treated with the blockmiR. In comparison to the previously tested antagomiR, the two strategies show a similar pattern of expression in many genes but with the antagomiR being more intense (see Figure 3b gold box). Indeed, antagomiR tended to show over-recovery in comparison to the control. In order to compare different DM1 models, RNASeq data was also retrieved from http://DMSeq.org (accessed on 13 November 2020). to analyze DM1 muscle biopsies [50]. DMSeq.org is a deep sequencing data repository for omics data from DM human samples and animal models. There is a noticeable difference between the DM1 cell model and the DM1 biopsies compared to the control. The DM1 transdifferentiated myotubes showed many differences while the biopsies show fewer changes in these 25 genes. Nevertheless, M1-Block-23b-1 treatment still rescued the transcript expression upregulated in the biopsies including *PODN*, *CLSTN2*, *RAI2*, and *MYH6*.

Interestingly, of all the genes showing total recovery by the blockmiR, 14 of these were over-represented as GOterms with relation to muscle and skeletal function (Figure 3c). To understand their connection, these genes were used as input to the STRING Protein-Protein Interaction Network database to find the most significant interaction linkages. Importantly, the myosin heavy chain (MYH) family of proteins and myogenin protein (MYOG) were strongly connected. These proteins play a role in signaling myogenic differentiation of myoblasts into mature skeletal muscle [51]. In fact, the upregulation of the MYH proteins is strongly linked to DM patient biopsies [52]. This associates M1-Block-23b-1 with the recovery of gene expression involved in critical neuromuscular mechanisms of the disease.

### 3.4. M1-Block-23b-1 Has Highly Specific Effects

The blockmiR strategy is built on the demonstrated effectiveness of antagomiR technology described in our previous studies [22,23,24]. Both blockmiRs and antagomiRs aim to recover MBNL protein function through the de-regulation of miR-23b and -218. However, the blockmiR strategy differs from antagomiRs by explicitly blocking the binding sites of miR-23b or miR-218 so as not to affect the endogenous amount of these miRNAs. In this way, the blockmiR approach is expected to have a more specific effect. To confirm this hypothesis, RNASeq was used to identify “disease-related genes” showing alterations in expression between DM1 cells and CNT cells (Figure 4a). Of these 2177 identified genes, AntagomiR-23b treatment showed alterations in 1057 of them while M1-Block-23b-1 treatment showed alterations in 166 in comparison to untreated DM1 cells. Of the 1057 genes altered by AntagomiR-23b treatment, 48.53% were disease-related reversals (see Figure 4a gold squares). As for the BlockmiR treatment alterations, 29.52% were disease-related improvements. These reversals suggest that the treatments are provoking a normal expression pattern but do not quantify by how much. Therefore, the percentage of rescue was analyzed for all disease-related genes and cross-referenced to the treatments (Figure 4b). When compared to CNT cells, M1-Block-23b-1 and AntagomiR-23b were capable of partial and total recovery. The BlockmiR showed greater partial recovery while the AntagomiR showed greater total recovery but even 108 events of over-recovery. The BlockmiR and AntagomiR also recovered genes that were exclusive from one another. Looking at Figure 4a,b, the AntagomiR has a more intense effect on the wider transcriptome.

In order to confirm the specificity of the two strategies, RNASeq data was used to analyze the transcripts targeted by miR-23b (identified using miRTarBase). For the BlockmiR-treated cells, only 1 of the 57 hard targets showed a change in expression in comparison to DM1 cells (see Figure 4c gold box). In contrast, there were 28 alterations in the AntagomiR-treated cells of the 57 targets. In comparison to healthy cells, M1-Block-23b-1 treatment showed more resemblance to the control than AntagomiR-23b treatment, indicating that targeting the miRNA binding sites is a more specific method of miRNA de-regulation than direct targeting of the miRNA itself.

### 3.5. BlockmiR Administration In Vivo Increases Mbnl1 Protein and Rescues Symptoms

There are two predicted binding sites for miR-23b on *Mbnl1* in mice. The first site (MMU-23b-Site1) is non-homologous with human *MBNL1*. The second binding site (MMU-23b-Site2) in mice has one nucleotide difference in comparison to the homologous human miR-23b site (HSA-23b-Site1) (Figure 5a). Therefore, a new blockmiR, named M1-Block23b-M, was generated considering the single nucleotide difference. M1-Block23b-M was administered through subcutaneous injection at 12.5 mg/kg in HSA^LR^ DM1 model mice (*n* = 5). The human version of the blockmiR, M1-Block23b-H, was used as a control and was also injected at the same concentration (*n* = 5). A group of FVB healthy mice (*n* = 5) and a group of HSA^LR^ mice (*n* = 5) were administered PBS as two separate controls. Four days after the initial injection, the mice were sacrificed, and their blood serum, gastrocnemius, and quadriceps muscles were extracted for further examination.

Biochemical analysis of blood serum was performed for each sample and compared to the PBS levels (Figure 5b). In most cases, M1-Block23b-H remained near the same level as PBS. However, significant decreases were seen for M1-Block23b-M in amylase and ALT. The acceptable ranges of biochemical parameters in blood differ between mouse strains [53]. To better understand the acceptable ranges of each measurement, a separate graph was included in Appendix A with empirical data from 12 male FVB mice between 12–18 weeks old from our lab. The average for each mouse was graphed along with the upper and lower standard deviation shown in green. The dashed grey line is the average of male FVB mice 6–8 weeks old [54] when data was available. With the results in this context, all the values that were affected by the M1-Block23b-M treatment were still found to be within or close to the known values of healthy FVB ranges.

The migration of muscle fiber nuclei towards the center of the muscle fibers is a typical feature of HSA^LR^ mouse muscle [32]. To observe this, hematoxylin and eosin (H&E) staining was performed on both gastrocnemius and quadriceps muscle, and the percentage of muscle fibers containing central nuclei was calculated (Figure 5c,d). In both tissues, M1-Block23b-M-treated cells showed a significant reduction in central nuclei compared to PBS controls. These fibers also show morphological similarity to healthy mice. Images were randomized so that nuclei could be quantified blindly to avoid bias. Detailed statistical information for the fiber analysis can be found in Appendix A. M1-Block23-M also increased the grip strength of DM1 mice (Figure 5e). The percent normal force was calculated by measuring the grip strength of each mouse before and after treatment, normalizing to their weight, and statistically comparing to PBS levels.

The *Mbnl1* transcripts were also significantly increased in gastrocnemius and quadriceps after M1-Block23b-M treatment (Figure 6a,b). This likewise corresponded to Mbnl1 protein levels after blockmiR administration (Figure 6c,d). Mbnl2 was also increased in the mouse muscle with the quadriceps showing higher rescue than gastrocnemius (Figure 6e,f). M1-Block23b-M treatment did not significantly increase Mbnl2 compared to PBS levels but compared to the human oligo counterpart, it is significantly higher. In all experiments, the M1-Block23b-H served as an excellent control showing no difference between the PBS treatment while being chemically very similar to the mouse-specific blockmiR by differing by only one nucleotide.

The splicing of four transcripts that show aberrant patterns in mice was analyzed for each treatment including *Atp2a1*, *Clcn1*, *Nfix*, and *Mbnl1* (Figure 6g–j). Small percent splicing recovery (PSR) could be seen for the averages of each treatment in exons in *Clcn1*, *Mbnl1*, and *Nfix*. But when observing the mice individually, two mice, in particular, showed strong PSR across all transcripts.

## 4. Discussion

The use of blockmiR miRNA site-blocking technology is a highly specific method of antisense oligo therapy. In this study, blockmiRs successfully block the binding of miRNAs in order to restore MBNL protein function in DM1 cells. The LNA-based oligos increased the expression of *MBNL* transcripts as well as corrected aberrant splicing patterns. Specifically, M1-Block-23b-1 and M2-Block-23b-1 showed rescue of *MBNL-1* and -*2* transcripts as well as *MBNL1* and *SPTAN1* mis-splicing. Importantly, *MBNL1* exon 5 inclusion is associated with the nuclear localization of MBNL1 proteins [55,56]. Indeed, the same blockmiRs with exon 5 rescue made a significant change in the intensity and subcellular localization of MBNL1 proteins in the cell. This is a critical therapeutic effect in DM1 treatment, as MBNL sequestration in the nucleus is the limiting factor in this disease. M1-Block-23b-1 specifically showed a dose-response increase of MBNL1 protein when treated at increasing concentrations.

Of interest in this study, M1-Block-23b-1, which targets an *MBNL1* miR-23b binding site, increased *MBNL1* and *MBNL2* transcripts. M2-Block-23b-1, targeting an *MBNL2* site, was also able to increase both transcripts. This could be due to the partial redundancy of the MBNL-1 and -2 proteins and regulatory feedback loops existing between the two [57]. In this way, rescue in one transcript could be expected in the other. This also could explain why slight rescue was seen in *SPTAN1* alternative splicing after treatment with M1-Block-23b-1 even when *SPTAN1* is regulated by MBNL2 proteins.

Curiously, the M2-Block-218-2 showed a large increase in MBNL1 pixel intensity without any *MBNL1/2* transcript increase or *MBNL1* splicing rescue. This could also be due to the MBNL protein functional overlap mentioned previously. Another hypothesis is that perhaps the de-regulation of *MBNL2* transcripts and subsequent increase in MBNL2 protein could increase competitive binding to the *DMPK* repetitions. In this way, it is possible that an increase in MBNL2 could trigger an increase in functional MBNL1 that has been released from *DMPK* repetitions.

It is also curious that though MBNL1 protein was increased after treatment with M2-Block-218-2, there was no discernable splicing rescue in *MBNL1* exon 5. Indeed, exon 5 inclusion is associated with the nuclear localization of MBNL1 proteins. However, the exclusive distribution of MBNL1 in the nucleus relies on the concurrent inclusion of exon 5 and exon 6 in *MBNL1* [56]. Therefore, even though the DM1 cells still showed exon 5 inclusion after treatment with M2-Block-218-2, exon 6 exclusion could be responsible for contributing to the cytoplasmic subcellular distribution. There is also evidence for the role of exon 7 in the effects on MBNL1 localization [10].

Another explanation for this splicing pattern could be due to the site-specific nature of the blockmiRs. Unlike antagomiR technology, blockmiRs only block binding sites of miRNAs, not the miRNAs themselves. In this way, the miRNA of interest is still free to bind to other target sites. In the context of this experiment, miR-218 could still bind to other binding sites on *MBNL2* while M2-Block-218-2 is bound and cause regulation of the transcript. In this way, repression of miR-218 on *MBNL2* could cause an overcompensation of miR-218 activity on other 3′-UTR binding sites. Perhaps blocking all miR-218 sites at one time would increase effectiveness such as the combinatorial usage of M2-Block-218-1 and -2, and -3.

M1-Block-23b-1 was chosen as the candidate compound after the preliminary screening due to its performance in cells and due to it targeting the only binding site for miR-23b in the human *MBNL1* 3′-UTR. Therefore, potential effects could be more easily interpreted. After performing RNA sequencing and looking at the top 25 genes rescued by M1-Block-23b-1 treatment, a network of important proteins involved in muscle function and infrastructure was revealed through STRING analysis. Most notably the myosin heavy chain (MYH) family of proteins and myogenin protein (MYOG) are critical for inducing myogenic differentiation of myoblasts into mature skeletal muscle [51]. Indeed, upregulation of the MYH family of proteins is strongly associated with DM patient biopsies [52].

When compared to the previously studied antagomiR-23b through RNASeq data, the M1-Block-23b-1 showed less dramatic gene expression recovery. Indeed, the antagomiR is a miRNA-specific technology while the blockmiRs are site-specific. However, there were no cases of blockmiR-induced over-recovery unlike with antagomir-treated cells where 108 genes were over-recovered. Finally, the blockmiR had no effect on other miR-23b targets whereas antagomiR-23b had many. This confirmed the hypothesis that the blockmiR strategy has highly specific effects on the cell transcriptome and merited further study in vivo.

M1-Block23b-M was able to make stunning increases in Mbnl-1 and -2 protein expression. MBNL1 is believed to play a larger role in cardiac and skeletal muscle while MBNL2 likely plays a larger role in the brain contributing to DM1 central nervous system (CNS) phenotypes [9,42]. The effects of the Mbnl1 rescue could be seen in mice with the drastic improvement of grip strength and skeletal muscle histology of the quadriceps and gastrocnemius muscles. However, the HSA^LR^ mice model does not exhibit cardiac or CNS phenotypes. Future experiments should include mouse models exhibiting these phenotypes such as DM500 or DMSXL mice [58,59] in order to see if the blockmiR can rescue other symptoms.

A few significant changes were observed when comparing M1-Block23b-M to FVB levels in the biochemical analysis. However, when these results are put in the context of the added reference FVB levels, some of the values that were changed are still within the known values of healthy FVB ranges such as alkaline phosphatase and ALT. Indeed, in some cases, M1-Block23-M caused favorable changes, such as lowering cholesterol, AST, ALT, and CPK [60,61,62,63]. The decreases in ALT and AST also caused the AST:ALT ratio to decrease. An AST:ALT ratio closer to 1 is considered healthy [64]. Additionally, CPK, cholesterol, and ALT have all been found to increase in DM2 [65]. Therefore, their decrease after treatment with M1-Block23b-M could be interpreted as favorable. When compared to all-LNA gapmers that showed high toxicity in hepatic biomarkers in CD-1 mice [66], the ALT, AST, and alkaline phosphatase levels of blockmiR treated mice were within the control values as well.

In splicing, the blockmiRs showed less consistent results in vivo than they did in vitro. However, two mice showed exceptional rescue of fetal splicing patterns in *Atp2a1*, *Clcn1*, *Mbnl1*, and *Nfix*. The rescue of these transcripts is of clinical importance to DM1. *ATP2A1* exon 22 exclusion is associated with muscle dysfunction, *CLCN1* exon 7a inclusion is associated with myotonia, and *MBNL1* exon 5 inclusion is associated with MBNL1 subcellular localization.

The modulation of Mbnl1 protein by the blockmiR is consistent with the results seen from the previously published P-PMO blockmiR as well as the antagomiR also by our group. Indeed, the blockmiR was able to increase Mbnl1 protein approximately 3.5-fold in the quadriceps and gastrocnemius. The P-PMO blockmiR showed increased Mbnl1 by 1.5-fold in quadriceps, albeit at a slightly lower concentration of 10 mg/kg. AntagomiR-23b in mice at the same concentration as the LNA blockmiR was able to increase Mbnl1 4-fold in quadriceps and 2-fold in gastrocnemius. A critical setback from AON technology is the ability to deliver the oligos to all tissues at the target concentration. M1-Block23b-M shows advantages over the P-PMO based blockmiR by showing therapeutic effects in both muscles. Another advantage of the blockmiR technology is its site-specific nature, which avoids undesired effects on the wider transcriptome. Further optimization of the oligo modifications could help solidify the therapeutic potential of the blockmiR strategy. Overall, the recovery of Muscleblind protein functionality in mice as well as highly specific effects on the cell transcriptome are edifying evidence for blockmiRs as restorative medicine for DM1 patients.

## 5. Patents

R.A., B.L., and E.C.-H. are co-inventors in patent PCT/EP2017/073685, currently licensed to Arthex Biotech, which includes blockmiR technology.

## Figures and Tables

**Figure 1 pharmaceutics-15-01118-f001:**
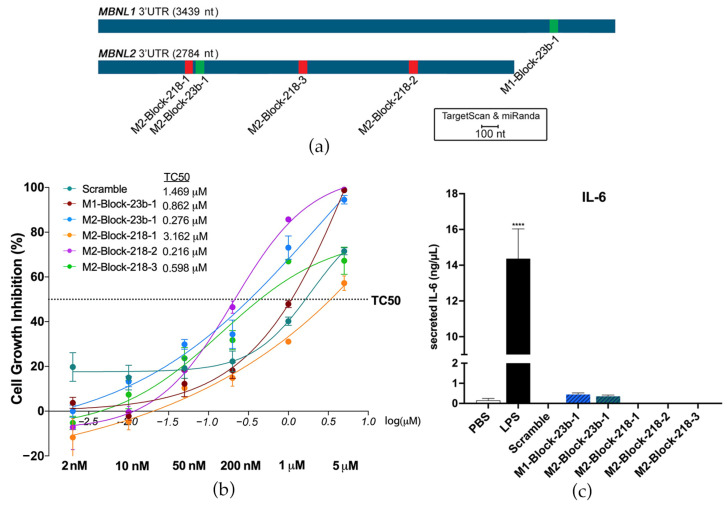
BlockmiRs and their effects on toxicity and immune response. (**a**) miRNA binding sites on *MBNL1* and *MBNL2* 3′-UTRs were predicted using TargetScan and miRanda and confirmed by dual luciferase assay. (**b**) One blockmiR was designed for each confirmed binding site and tested for cell viability at six concentrations (shown underneath the *x*-axis in bold) in DM1 cells. Data are shown as the average of four technical replicates log-transformed (*x*-axis) with the dotted horizontal line showing the threshold concentration at 50% (TC_50_). The untransformed TC50 for each compound can be seen on the table to the right of the legend. (**c**) Immune activation by blockmiRs in C2C12-derived 3D mouse muscle tissues was determined through IL-6 quantification with three technical replicates. BlockmiRs were all administered at 500 nM. All samples were compared to the Scramble via one-way ANOVA. *p*-value: *p* ≤ 0.0001 (****). Error bars = SEM. Reproduced/adapted from [29] Universitat de València, 2022.

**Figure 2 pharmaceutics-15-01118-f002:**
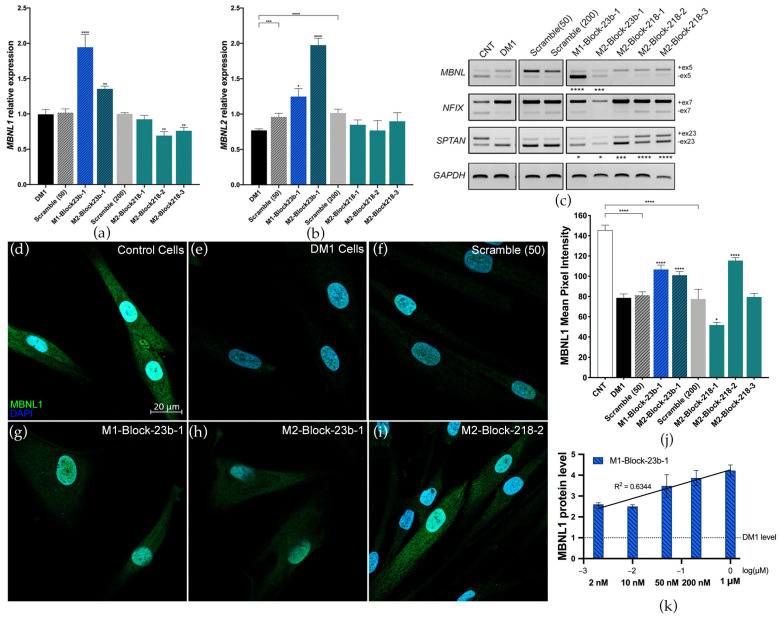
BlockmiRs increase MBNL and improve mis-splicing. Relative expression of (**a**) *MBNL1* and (**b**) *MBNL2* transcripts after treatment with blockmiRs through RT-qPCR. Data are shown as the average of four technical replicates. *GAPDH* was used as an endogenous control and MBNL levels were calibrated relative to the respective Scrambles (50 or 200). (**c**) Alternative splicing was analyzed for transcripts *MBNL1*, *NFIX*, and *SPTAN1* with *GAPDH* used as an endogenous control. A representative gel is seen above. Amplicons were generated using three different cDNA replicates and averaged for statistical analysis using Image J. (**d**–**i**) CNT and DM1 cells were immunostained in parallel for MBNL1 protein (green) and nuclei (DAPI blue). Healthy control cells (**d**) show intense and localized MBNL1 fluorescence in the nucleus and cytoplasm. DM1 cells (**e**) show only punctate fluorescence of MBNL1 in nuclear foci. (Cell count: CNT *n* = 63; DM1 *n* = 106; Scramble 50 *n* = 105; M1-Block-23b-1 *n* = 86; M2-Block-23b-1 *n* = 145; M2-Block-218-2 *n* = 155) (**j**) MBNL1 protein fluorescence was quantified by measuring the mean pixel intensity (Threshold = 10) using Image J normalized by cell area. (**k**) MBNL1 expression in DM1 cells after treatment with M1-Block-23b-1 at increasing concentrations was analyzed using quantitative dot blot (QDB) analysis. All samples were calibrated with GAPDH levels. Data are shown as the average of four technical replicates log-transformed and normalized relative to untreated DM1 cell MBNL1 levels. The black line shows the linear regression with the R^2^ value seen above. All samples were compared to their respective Scrambles (50 or 200) via Student’s *t*-test. *p*-value: *p* ≤ 0.05 (*), *p* ≤ 0.01 (**), *p* ≤ 0.001 (***), *p* ≤ 0.0001 (****). Error bars = SEM. Reproduced/adapted from [29] Universitat de València, 2022.

**Figure 3 pharmaceutics-15-01118-f003:**
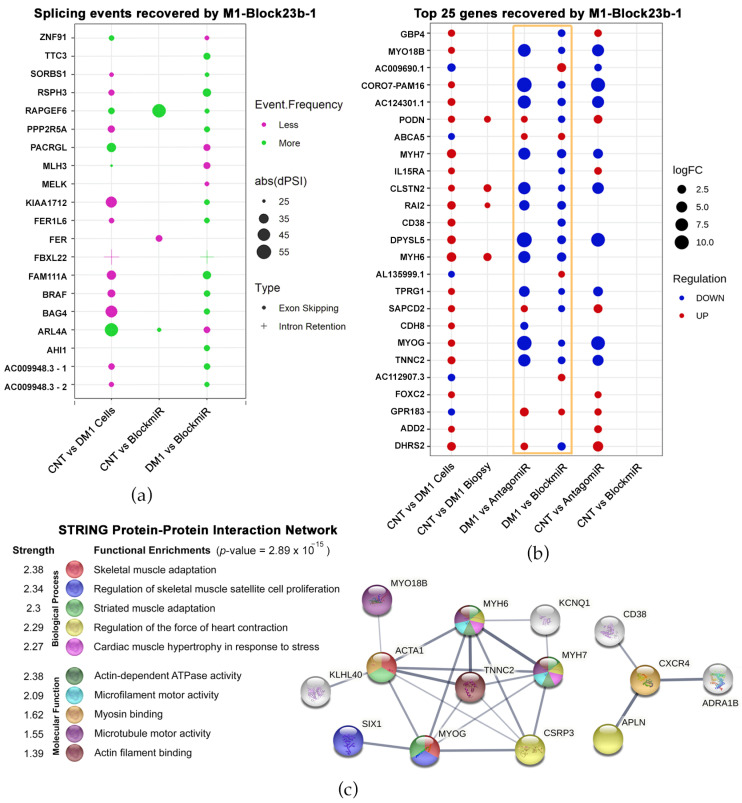
Transcriptomic recovery of blockmiRs through RNASeq analysis. (**a**) Splicing events were identified at exon/intron junctions and quantified using delta percent spliced-in (dPSI). Events were filtered for greater than 25% dPSI (represented by geometric point size), adjusted *p*-value < 0.05, and >10 reads. (**b**) Geometric points are plotted for the top 25 genes that showed total recovery after treatment with the blockmiR. Point size represents the strength of the log2 fold change, while the color represents the under- or over-expression of each comparison. (**c**) The 135 genes with total recovery by the blockmiR were analyzed with GOterm with 14 edges found over-represented. These edges were searched in the STRING Protein-Protein Interaction Network database to find both functional and physical protein associations. Line thickness reflects the strength of supporting data. Reproduced/adapted from [29] Universitat de València, 2022.

**Figure 4 pharmaceutics-15-01118-f004:**
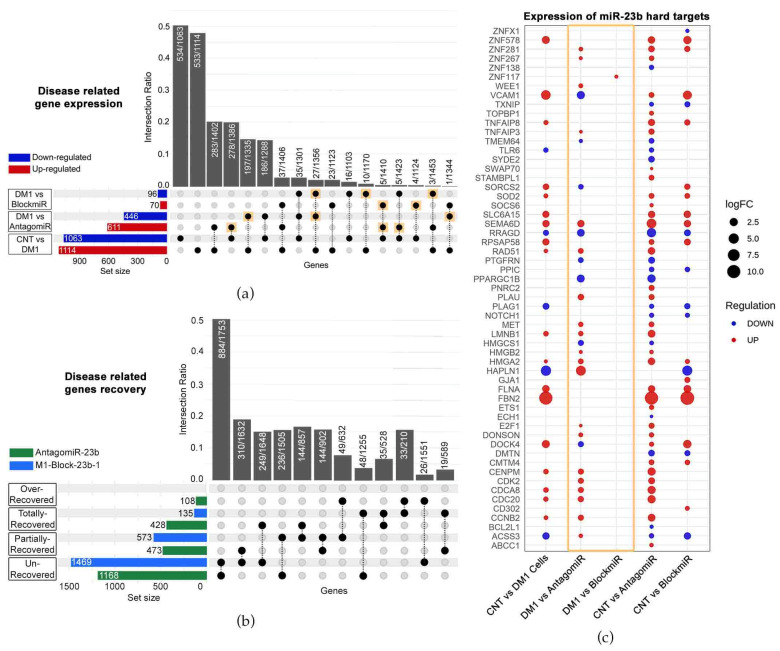
The transcriptomes of BlockmiR- vs. AntagomiR-treated DM1 cells. (**a**) When comparing CNT to DM1 cells, 2177 altered genes were labeled disease-related genes. A total of 1057 genes and 166 genes were altered after AntagomiR-23b (AntagomiR) and M1-Block-23b-1 (BlockmiR) treatment, respectively, in comparison to untreated DM1 cells. These numbers can be seen by summing the number of genes up- or down-regulated in the respective red and blue horizontal bars to the left of each comparison. The black circles and dotted lines show which genes are common to each group. The number of these shared genes divided by the total number of unaffected genes from the groups is calculated as the intersection ratio. (**b**) The number of genes with recovery of disease-related genes can be seen for both AntagomiR (green bar) and BlockmiR (blue bar). Unrecovered < 10%; partially recovered between 10% and 50%; total recovery between 50% and 150%; over recovered > 150%. The black circles and dotted lines show which are common to each group. The number of these shared genes divided by the total number of unaffected genes in each group is calculated as the intersection ratio. (**c**) Geometric points are plotted to represent the log2 fold change of the hard targets of miR-23b after treatment with the BlockmiR or AntagomiR. The size of each point reflects the intensity while the color reflects the over- or under-expression of each comparison. Reproduced/adapted from [29] Universitat de València, 2022.

**Figure 5 pharmaceutics-15-01118-f005:**
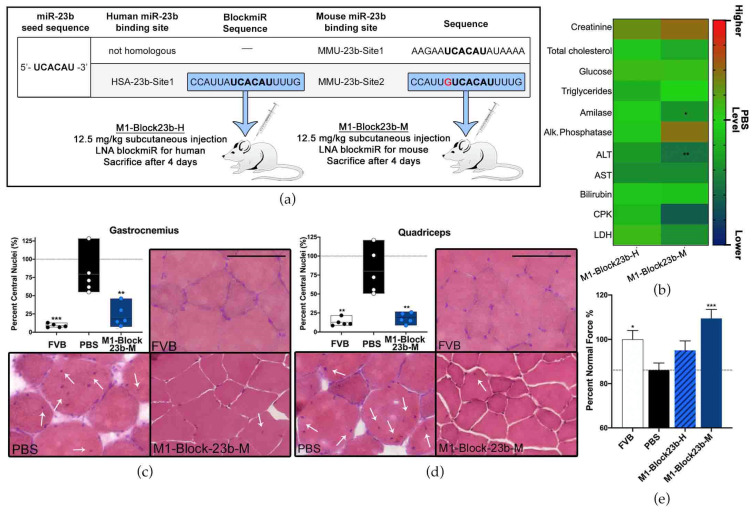
In vivo blockmiR administration is followed by physiological improvements. (**a**) The sequence for the miR-23b binding site targeted in human DM1 cells has one nucleotide difference (red) for the homologous site in mice. Therefore, a new blockmiR was created to account for this single nucleotide difference to facilitate specific binding in HSA^LR^ mice. Both the human and mouse blockmiRs were injected at 12.5 mg/kg into HSA^LR^ mice. HSA^LR^ mice given PBS and healthy FVB mice were used as controls. FVB (*n* = 5), PBS (*n* = 5), M1-Block23b-H (*n* = 5), M1-Block23b-M (*n* = 5). (**b**) Toxicological parameters were assessed after treatment and the raw values can be found in Appendix A. PBS levels are shown in green. Values above PBS increase to red and values below PBS decrease to blue. All samples are compared to PBS through one-way Kruskal–Wallis ANOVA. (**c**) Gastrocnemius and (**d**) quadricep muscle were cut into 10 μm sections and stained with hematoxylin and eosin in order to observe the position of nuclei within the muscle fibers. For each mouse, an average of 600 fibers was identified and counted. Representative images of the muscles from each treatment can be seen. Black scale bar = 100 μm. White arrows indicate nuclei with central localization. (**e**) Mouse grip strength was measured three times both before and after treatment with M1-Block23b-M by grip dynamometer to calculate the percent of normal force normalized to mouse weight. All samples were statistically compared to PBS through Student’s *t*-test. *p*-value: *p* ≤ 0.05 (*), *p* ≤ 0.01 (**), *p* ≤ 0.001 (***). Error bars = SEM. SEM and number of fibers for panels (**c**,**d**) can be found in Appendix A. Reproduced/adapted from [29] Universitat de València, 2022.

**Figure 6 pharmaceutics-15-01118-f006:**
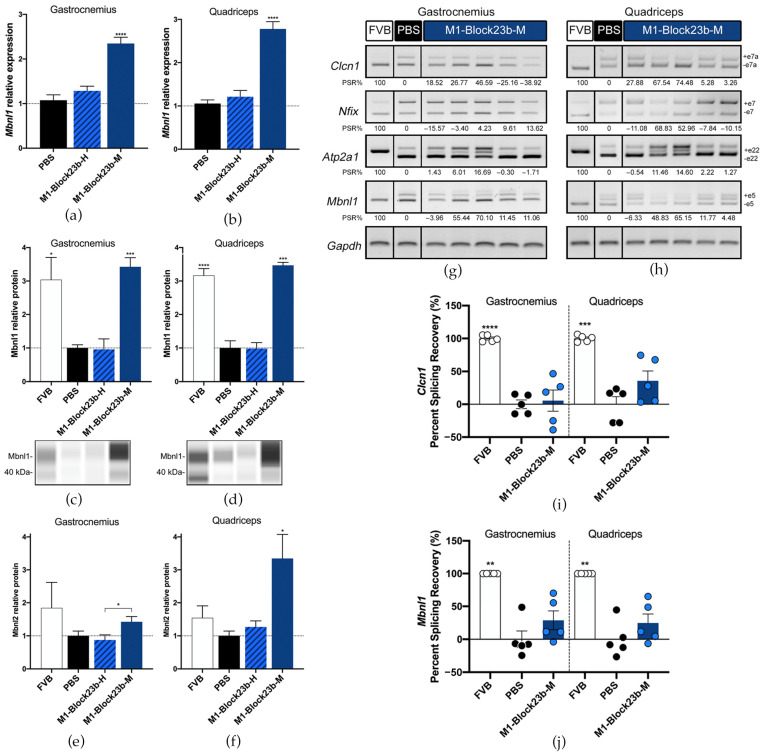
Molecular rescue by blockmiR in mice. Relative expression of *Mbnl1* transcripts in (**a**) gastrocnemius and (**b**) quadriceps muscles was relatively quantified with three technical replicates from each mouse after treatment with the blockmiRs. The data were normalized to *Gapdh* as an endogenous control. (**c**,**d**) Jess Simple Western technology was used to relatively quantify Mbnl1 protein for each muscle. Bands from a representative lane view can be seen below each graph. (**e**,**f**) Mbnl2 was quantified through sandwich ELISA and the average of three technical replicates was normalized to total protein from BCA quantification. Alternative splicing was analyzed for transcripts *Atp2a1*, *Clcn1*, *Mbnl1*, and *Nfix* with *Gapdh* used as an endogenous control. A representative gel is seen above for (**g**) gastrocnemius and (**h**) quadriceps. Amplicons were generated using three different cDNA replicates and averaged for statistical analysis using Image J. The quantification can be seen in the percent splicing recovery (PSR%) noted below each band. The same data represented in a scatter plot is presented in panel (**i**) for *Clcn1* and (**j**) for *Mbnl1*. All statistical comparisons were performed against PBS via Student’s t-test. *p*-value: *p* ≤ 0.05 (*), *p* ≤ 0.01 (**), *p* ≤ 0.001 (***), *p* ≤ 0.0001 (****). Error bars = SEM. Reproduced/adapted from [29] Universitat de València, 2022.

**Table 1 pharmaceutics-15-01118-t001:** BlockmiR design.

Target	Modified Sequence	Concentration
M1-Block-23b-1	CbsCbsAbsTbsTbsAbsuscsascsasusususTbsGb	50 nM
M2-Block-23b-1	AbsuscsascsasusgsasTbsTbsCbsAbsAbsCbsGb	50 nM
M1-Block-218-1	GbsAbsusgsusgscsusususAbsAbsAbsTbsAbsTb	200 nM
M1-Block-218-2	GbsususgsusgscsusgsTbsCbsTbsAbsTbsTbsGb	200 nM
M2-Block-218-1	AbsCbsTbsusgsusgscsususGbsAbsAbsusTbsTb	200 nM
M2-Block-218-2	GbsTbsTbsGbsusgsusgscsusasasTbsAbsAbsTb	200 nM
M2-Block-218-3	CbsGbsAbsTbsAbsgsusgscsususAbsAbsAbsAb	200 nM
Scramble	TbsGbscsascscsusususgsTbsTbsAbsTbsTbsTb	50 nM & 200 nM
M1-Block23b-H	CbsCbsAbsTbsTbsAbsuscsascsasusususTbsGb	12.5 mg/kg
M1-Block23b-M	CbsCbsAbsTbsTbsGbsuscsascsasusususTbsGb	12.5 mg/kg

Legend: 2′-O-methyl nucleotides = lower-case a, g, c, u; LNA nucleotides = Ab, Gb, Tb, Cb; Phosphorothioate linkages = s.

## Data Availability

Data deposition: RNA-sequencing data have been deposited in the Gene Expression Omnibus (GEO) database, https://www.ncbi.nlm.nih.gov/geo (accession no. GSE173359, accessed on 26 April 2021).

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
