# Peer review of "BlockmiR AONs as Site-Specific Therapeutic MBNL Modulation in Myotonic Dystrophy 2D and 3D Muscle Cells and HSALR Mice"

_pharmaceutics, 2023, doi:10.3390/pharmaceutics15041118_

Round 1
Reviewer 1 Report
In my opinion, the authors should clearly state what this study adds to the previously published, nearly identical, study (pubmed.ncbi.nlm.nih.gov/35282418/).
The novelty of the same strategy using a different chemistry should be explained better and explicitly stated in the manuscript. Additionally, the comparison of efficiency between the P-PMO and LNA should be presented more clearly side to side.
Reviewer 2 Report
The manuscript entitled "BlockmiR AONs as site-specific therapeutic MBNL modulation in Myotonic Dystrophy cells and mice" has been carefully reviewed. I've found it a very well-written manuscript that presented promising results in the miRNA-based therapeutic approach. The study was conducted in a scientific manner which led to achieving reliable data. I believe that this study will be one of the referenced ones in the field due to the extensive and detailed evaluations. I recommend this manuscript to be published in Pharmaceutics in its current form.
Reviewer 3 Report
i. Consider adding a small graphical diagram to the Introduction to visualize the differences between blockmiR strategy and antagomiR strategy.
ii. Consider moving the table with the sequences from the supplementary information to the main text, possibly with the structures of the used modifications for better orientation of the reader.
Reviewer 4 Report
1. Title: Using acronyms does not help the paper. Perhaps, consider, "BlockmiRNA antisense oligonucleotides target site-specific muscleblind like splicing regulator 1 (MBNL1) in therapeutic treatment of myotonic dystrophy type 1 (DM1) 2D and 3D muscle cells and in homozygous transgenic HSALR (line 20 b) mice"
2. Abstract: please use the precise definitions of terms, For example, MBNL is muscleblind like splicing regulator.
3. Mouse 3D skeletal muscle tissues biofabrication: please show images of the 3D muscle tissues along with the data.
4. Cell transfection: Cells were plated at 1.0x105 cells per mL in 10 mL petri dishes. What cells were used here? Fibroblasts or muscle? Please be precise. Also, A scrambled control (Scramble) was transfected at 50 nM and 200 nM? What was used here?
5. To measure cell viability, 20 μl of MTS/PMS tetrazolium salt from the CellTiter 96 Aqueous Non-Radioactive Cell Proliferation Assay: Please show the formula to calculate cell viability taking into account the media with treatment agents.
6. Lipopolysaccharide (LPS) at 10 μg mL-1 (3 replicates per assay). Have you done a dose-dependency of IL6 of internal positive control.
7. All figure legends must indicate how many independent experiments were performed. Triplicate in one assay is not appropriate.
8. Fig 1b - not clear from the figure what is log(x) transformed of what and the threshold concentration values at 50% (TC50) for each compound? How were TC50 values performed?
9. Fig1c - it is not clear if some of the results were zero or not. The Y-axis should be split and extend or enlarge any values from 0 to 1. The Y-axis should be labelled proper using IL6 secreted (ng/mL).
10. Fig 2c: A representative gel is seen above. These blots were performed by the Jess Western technology. Authors must provide the full length blots as supplementary data before any consideration for publication. The same issue with Figure 6 blots.
11.
Round 2
Reviewer 1 Report
I still feel that the manuscript is well-written and the data is presented well.
I feel the experiments lack the previously published P-PMO as control. The authors compare the two compounds on page 19 line 732-738. The P-PMO should be included as control in the experiments. The authors should state the molar mass differences as well. The P-PMO was administered at 10mg/kg, and the LNA comprising oligo was administered at 12,5 mg/kg, how comparable is this on a molar level? Because 25% does not sound that small of a difference.
If the PMO is not directly compared to the LNA chemistry in the same experiments. Therefore, all statements with regards to comparing the two compounds are speculative. This should be mentioned.
Reviewer 4 Report
The authors responded adequately to peer reviewers' comments.